# Interference Analysis of 5G NR Base Stations to Fixed Satellite Service Bent-Pipe Transponders in the 6425–7125 MHz Frequency Band

**DOI:** 10.3390/s23010172

**Published:** 2022-12-24

**Authors:** Alexander Pastukh, Valery Tikhvinskiy, Evgeny Devyatkin, Artyom Kostin

**Affiliations:** 1Radio Research and Development Institute, 105064 Moscow, Russia; 2International Information Technologies University (IITU), Almaty 050000, Kazakhstan; 3Geyser-Telecom Ltd., 105118 Moscow, Russia

**Keywords:** 5G, Monte-Carlo analysis, fixed satellite service, bent-piper transponder, 6425–7125 MHz, interference analysis, frequency management, spectrum sharing, 5G digital beamforming, BER

## Abstract

Future deployment of 5G NR base stations in the 6425–7125 MHz band raises numerous concerns over the long-term impact on the satellite transponders located in geostationary orbit. To study this impact and understand whether 5G NR may cause adverse effect to the spaceborne receivers, the research which estimated the interference levels to the satellite bent pipe links was done. The study presents the evaluation of aggregate interference from 5G NR base stations located inside the victim satellites’ footprints using Monte-Carlo analysis and calculation of signal-to-noise degradation and bit error rates of the fixed-satellite service (FSS) bent-pipe transponders for each scenario. The results of the study showed the feasibility of co-existence between 5G NR and satellite systems in the 6425–7125 MHz bands, and that no negative impact on the performance of the satellite links is expected.

## 1. Introduction

This work provides extended research to the study [1] related to sharing of the frequency band 6425–7125 MHz, between 5G and fixed satellite services transponders in geostationary orbit. In the previous study, only one satellite transponder was considered that covered the Eastern European region, whereas in this study two additional transponders that cover the Eurasian landmass and Pacific region are considered. Additionally, the authors greatly extended scenarios of calculation. First, apart from urban and suburban deployment of the interfering 5G base stations, rural deployment was taken into account as well, second in the calculation, several new modulation and coding rates of the victim receiver modems were considered which allows for more better understanding how 5G deployment will affect the performance of satellite bent pipe transponder links. 

Governments of many states around the world recognize the importance of the 5G mobile communication services and their evolution as one of the main drivers in the development of their economies. In the past, cellular technologies provided voice and Internet access services. Now with the advent of 5G, the set of provides services has expanded to many other applications such as immersive communications (AR/VR), smart cities, the smart industry with low latency, smart deliveries with drones, robotics, and many others. These new usage scenarios will lead to traffic volumes that will surpass the smartphones’ traffic volumes. Recent studies concluded that 5G will increase world GDP by 960 billion dollars by 2030, 610 billion of which will be the result of the 5G rollout in the mid bands. However, it is also concluded that this forecast won’t come true without allocating additional mid-band spectrum for mobile services.

5G networks utilize bands below 6 GHz (FR1) which have good coverage but lower data rates and the bands above 24 GHz (FR2) which quite contrary, have high data rates but poor coverage area. The 6425–7125 MHz frequency band provides a perfect compromise between the data rates of FR2 and the coverage areas of FR1. Extension of the 5G standard to the 6425–7125 MHz will allow increasing the effectiveness of 5G networks.

To realize 5G in the 6425–7125 MHz band it is required to do sharing and co-existence studies with incumbent services in this band. In the Radiocommunication sector of the International Telecommunications Union (ITU-R) under agenda item 1.2 of World Radiocommunication Conference 2023 (WRC-23) the question of identification of the 6245–7125 MHz band to International Mobile Services (IMT) is studied. The 6425–7125 MHz frequency band is allocated to the Fixed Satellite Service (Earth-to-space) and is utilized by the satellite transponders in geostationary orbit to provide communication services. Thus, before rolling out 5G networks, it is important to do interference analysis studies and figure out whether satellite links may experience disruptive interference levels from 5G.

## 2. State of the Art

The 6425–7125 MHz frequency band is studied for possible 5G usage in ITU for several years, several works devoted to this band were published recently, however, they were related to compatibility with other incumbent services such as compatibility between 5G and feeder links of mobile satellite service in the frequency band 6700–7075 MHz [2] and with microwave stations operating as fixed service links [3,4]. Studies with fixed satellite service (Earth-to-space) for this band were not published yet. Similar studies were done for other bands such as 26 GHz [5,6,7,8], however, these studies considered compatibility between inter-satellite service where the space-to-space link is considered which has some significant difference between the type of the transmitted traffic and requirements to the radio link. Additionally, studies for 26 GHz considered the interference-to-noise protection criterion (I/N) which does not consider wanted signal links. While I/N is a pretty common protection criterion for satellite services that are used in ITU-R studies, this criterion is pretty stringent and does not reflect the real performance of the victim system. Using I/N quite often may lead to exaggerating impact levels which in practice will not occur. This study proposes to consider adaptive modulation and coding of the satellite links based on the DVB-S2 standard. This helps to do more precise studies estimating signal-to-noise degradation C/(N+I) of the satellite bent-pipe link and calculating the bit error rate (BER) for each modulation and coding scheme. This approach proposed by the authors is more realistic and allows to understand the real sharing conditions without imposing useless restrictions on the developing services.

## 3. Interference Scenarios and Parameters of Simulated Systems

The satellite provides an area coverage beam called the “footprint”, where the Earth-to-space link can originate from any location and the space-to-Earth link can likewise be received at any covered location. The uplink footprint would have contours measured in either saturation flux density (SFD) or gain-to-noise temperature ratio (G/T) [9]. Given that GSO satellites cover large areas, all base stations of 5G located inside the satellite’s footprint may cause interference, given that the large areas include hundreds of thousands of base stations, aggregate interference may reach significant levels. Figure 1 shows an example of possible interference from deployed 5G networks to the satellite bent pipe communication link. 

In the scenario shown above it may be seen that 5G networks may cause aggregate interference to the space-borne satellite receivers, taking into account that bent pipe links use transparent transponders where one Earth station transmits a signal to the satellite receiver at one frequency, then the onboard transponder translates the received signal to the other frequency, amplifies it and sends it to the other receiving Earth station. Transparent transponders are cheaper to realize, however, the cost of link degradation, since the interference received will be also amplified and retransmitted to the receiving Earth station causing increased bit error rate levels. In a typical 6/4 GHz transponder, there is no onboard processing done, and all signal degradations and noise received from the uplink are injected into the downlink together with the wanted signal, and thus the overall performance of the system will be dependent on both links [10]. Figure 2 below shows the simplified scheme of the transponder for the 6/4 GHz band with a single conversion:

First, the signal is received at the 6 GHz band and sent to a low noise amplifier which performs amplification of the received signal. Then the signal is down-converted to 4 GHz using a local oscillator with the 2.225 GHz frequency. After that, the intermediate frequency band pass filter (BPF) removes the undesired signals and a 4 GHz signal is sent to the pre-amplifier which consists of a traveling wave tube (TWT) and is fed to a high-power amplifier (HPA).

Three carrier satellites of Yamal and Express with orbital positions of 90E, 140E, and 183E were considered in the simulations as victim receivers. Table 1 provides the characteristics of each satellite link that was used in the simulations, these characteristics were derived from the Master Information Frequency Register database of the ITU Radiocommunication Bureau.

The study considers worst-case scenarios. At first glance, it may be suggested that the worst case would be for the footprints closer to the equator so the distance between satellites and 5G base stations would be shorter. This would be true only for the interference source with an omnidirectional antenna, while for the interferers with directional antennas this is not the case. Given that 5G base stations use beamforming, the amount of interference contribution in the Earth-to-space scenario depends more on the elevation angles towards the GSO arc rather than the distance, the lower the elevation angles, the higher interference the satellites will receive. This is why the northern parts of the Eurasian continent are more suitable since in such a geometrical configuration there would be cases when the main lobes of the BS beams will be pointed toward the GSO arc. Figure 3 presents the footprints of each satellite, the pink contour is a Yamal 90E footprint, the blue contour is an Express 140E footprint and the yellow contour is a Yamal 183E footprint.

Currently, there is no commercial equipment of 5G in the 6425–7125 MHz band, however, 3GPP and ITU-R provided characteristics of the base stations in this band which can be found in Document 5D/716 Annex 4.4. of ITU-R and in 3GPP TR 38.921 (2021-03) specifications. Table 2 provides characteristics of 5G BS that are used in simulations.

Among the enhancements of 5G NR compared to LTE is using beamforming technology, beamforming allows do electronic steering the direction of the main beam of the antenna pattern. The antenna patterns of 5G NR BS are presented in Figure 4 and Figure 5.

In simulations for each BS three multiple spatially directive signals are transmitted simultaneously in the direction of different UEs that are randomly distributed in the cell using a uniform distribution. The UEs which are served by each base station in the vertical coverage range should be considered to be served by the beam steered towards the maximum coverage angle, i.e., by the lower bound of the electrical beam. The minimum distance of UEs to each BS for urban, suburban, and rural deployments is 35 m. The antenna height of each UE is 1.5 m.

The conducted power per element assumes 16 × 8 × 2 elements (power per H/V polarized element). To calculate the total transmitted power emitted towards the direction of the victim receiver’s satellite, equivalent isotropic radiated power should be calculated using the following expression:EIRP_BS_ = P_tx_ + G_tx_
where P_tx_ is a conducted power of a 5G base station (dBm), G_tx_ gain towards the victim receiver (dBi), as expressed in Table 2 P_tx_ equals 46 dBm, whereas the gain towards the victim receiver depends on the antenna pattern presented in Figure 4 and Figure 5. Using that data, EIRP to each direction in azimuth and elevation plane can be considered. Figure 6 shows EIRP values in different directions when the antenna pattern is pointed toward the horizon. While scanning in the coverage area, the EIRP levels from each BS will vary in time and thus the interference levels will fluctuate depending on the beam direction of each BS. It should be noted that modern 5G BS uses digital beamforming. The major difference between analog and digital beamforming is that in digital beamforming multiple independent beams steered in all directions can be formed in the digital beamforming processor [11,12,13,14].

It is not expected for the 6425–7125 MHz range to be used in rural scenarios for providing contiguous coverage, however, some exceptions can be done and, and in such cases, base stations deployed in the rural area most likely will be isolated installations at some specific locations. When the rural deployment scenario is simulated in a sharing study, it should assume the BS density (per sector) of 0.001–0.006 BS per km^2^. Other parameters for the rural deployment should be the same as the suburban.

## 4. Simulation Methodology

The simulations were done using MATLAB and STK platforms. The methodology used in this study uses Monte-Carlo simulation to calculate interference level from IMT terrestrial networks to the GSO FSS uplink operating in the 6425–7125 MHz frequency band and then estimating degradation of the composite link taking into account downlink operation in the frequency band 3400–4200 MHz. One of the most challenging things in estimating interference from large areas of terrestrial networks towards GSO satellites is to understand the deployment density of terrestrial interfering stations inside the satellite footprint. ITU-R for that purpose developed a methodology to determine the density of the interfering stations. For the implementation of the methodology two parameters were developed, the first one is Ra which is the ratio of coverage areas to areas of cities/built areas/districts, the second is Rb which is the ratio of built areas to the total area of the region in study. The values of Ra and Rb depend on the region size and are presented in Table 3.

Total number of the 5G base stations inside the footprint may be calculated using the following expression:Dl_ur/sub_ = Ds _ur/sub_ ∗ Ra _ur/sub_ ∗ Rb (1)
where Ds is a density of simultaneously active BS transmitters per km^2^ for the reviewed coverage area, this value is provided in Table 2; Ra (%) is a ratio of coverage areas to areas of cities/built areas/districts; Rb (%) is a ratio of built areas to the total area of the region in study.

Given that this study involved large areas where mixed environments of urban, suburban, and rural are involved inside a satellite footprint, it may not be appropriate to assume that 5G base stations will be deployed with the same density as shown above across the entire footprint area. Thus, the deployment density values may need to be adjusted. This adjustment should be justified by the results of studies, e.g., by providing population density data and assumptions on coverage in less populated areas using the considered band.

It is proposed to quantify such an assumption with the factor Rc, described below. To consider the deployment of the rural case, factor Rc could be defined as the ratio of coverage areas to the areas in a rural environment. This factor could be applied in the following equation:Dl_ru_ = Ds_ru_ ∗ Rc ∗ (100-Rb)(2)

The total number of BS on the surface of the study is then based on the association of each Dl for each environment:Dl_tot_ = Dl_ur_ + Dl_sub_ + Dl_ru_(3)

Finally, the total number of BS in the footprint could be expressed as:Dl = (Ds_ur_ ∗ Ra_ur_ + Ds_sub_ ∗ Ra_sub_) ∗ Rb + Ds_ru_ ∗ Rc ∗ (100-Rb)(4)
where Ds _ur/sub_ is a density of simultaneously active BS transmitters per km^2^ for the reviewed coverage area with urban and suburban deployment; Ra_ur/sub_ (%) is a ratio of coverage areas to areas of cities/built areas/districts in urban (ur) or suburban (sub); Rb (%) is a ratio of built areas to the total area of the region in the study; Rc (%) is a ratio of coverage areas to rural areas.

By definition Rc (%) is a ratio of coverage areas to rural areas:Rc (%) = S_coverage_/S_rural_(5)

Ra (%) is a ratio of coverage areas to areas of cities/built areas/districts or
Ra (%) = S_coverage_/S_built-in_(6)

From (2): S_coverage_ = Ra (%) ∗ S_built-in_(7)

Substituting (7) to (5):Rc (%) = Ra (%) ∗ S_built-in_/S_rural_(8)

Thus, Rc is defined as a composite factor, already including the Ra assumption.

For the case of one particular Administration based on official statistics, these areas are S_built-in_ =84,004 km^2^ and S_rural_ = 120,526 km^2^. Considering Ra (%) = 30% Urban (area < 200,000 km^2^) the ratio of coverage areas to rural areas (adjustment factor) for this particular case can be expressed as follows:Rc (%) = Ra (%) ∗ S_built-in_/S_rural_ = 21% (9)

Therefore, the total number of BS in the footprint in the rural area could be found as Dl_ru_ = Ds_ru_ ∗ Rc ∗ (100-Rb), where Ds_ru_ = 0.001…0.006 BS per km^2^.

To calculate propagation losses between each interfering BS and victim the propagation model based on Recommendation ITU-R P.619 was used, the model takes into account losses in atmospheric gas, spreading losses, rain losses, etc. [15]. Additionally, the model based on Recommendation ITU-R P.2108 [16] was used which allows calculating slant path clutter attenuation, this mode is applicable for the frequency bands from 10 to 100 GHz, however, it can be extrapolated to the lower frequency bands, this extrapolation was approved by the ITU-R study group that is responsible for developing propagation models. In the study, clutter is applied for 100% BS with urban deployment, however, it wasn’t applied to the BS with suburban and rural deployment since the clutter height in suburban and rural areas will be lower than BS antenna heights according to the typical clutter classes in Recommendation ITU-R P.452.

To calculate *C/N* degradation, external noise addition expressed as *C/(N+I)* ratio should be taken into account. *C(N+I)* can be calculated using the following expression [15,16].
(10)$$C/\left(N+I\right)=C-10*\log\left(10^{\frac{I}{10}}+10^{\frac{N}{10}}\right)\;$$
where *N* is the noise level in the badnwidth of the receiver (dBW); *C* is the wanted signal level in the bandwidth of the receiver(dBW); I is the interference level in the victim receiver bandwidth (dBW).

It should be noted that since FSS links in the 6425–7125 MHz use transparent transponders and therefore there is no on-board processing or enhancement of the information signal, the composite *C/(N+I)* should be calculated. Composite *C/(N+I)* includes *C/(N+I)* levels in the uplink and downlink of the transponder link and should be calculated using the following expression [17,18]:(11)$$C/{\left(N+I\right)}_{total}=-10\log\left(10^{-0.1C/{\left(N+I\right)}_{up}}+10^{-0.1C/{\left(N+I\right)}_{down}}\right)$$
where *C/(N + I)_up_* is *C/(N + I)* of Earth-to-space transponder link (dBW); *C/(N + I)_down_* is *C/(N + I)* of space-to-Earth transponder link (dBW);

Scenario 1 is a beam that covers the Pacific region. 5G BS stations deployed at the footprint of the victim FSS satellite at orbital position 183° E. The transmitting ES of the bent pipe link is located at 47.49° N and 116.24° E and the ES that receives retransmitted signal from the satellite is located at 67.68° N and 152.84° E coordinates. The bent pipe link is shown in Figure 7. The satellite’s footprint area is 12,363,048 km^2^, whereas the landmass area of study excluding seas is 6,821,361 km^2^. The transmitting and receiving Earth stations are intentionally placed at the boundaries of the satellite footprint to consider the worst-case scenario and the bent pipe communication link is calculated, the 6 GHz uplink C/N = 15.9407 dB, the 4 GHz downlink C/N = 15.8195 and composite C/N = 12.8694 dB.

Scenario 2 is a global beam that covers parts of Eurasia. 5G BS stations deployed at the footprint of the victim FSS satellite at orbital position 183° E. The transmitting ES of the bent pipe link is located at 55.95° N and 68° E and the ES that receives retransmitted signal from the satellite is located at 66.85° N and 176.88° E coordinates. The bent pipe link is shown in Figure 8. The satellite’s footprint area is 67,237,138 km^2^, whereas the landmass area of study excluding seas is 30,305,000 km^2^. The transmitting and receiving Earth stations are intentionally placed at the boundaries of the satellite footprint to consider the worst-case scenario and the bent pipe communication link is calculated, the 6 GHz uplink C/N = 15.7588 dB, the 4 GHz downlink C/N = 17.4321 and composite C/N = 13.5051 dB.

Scenario 3 is a continental beam that covers Eastern European parts and some parts of Central Asia. 5G BS stations deployed at the footprint of the victim FSS satellite at orbital position 90°E. The transmitting ES of the bent pipe link is located at 53.38°N and 36.19°E and the ES that receives retransmitted signal from the satellite is located at 66.68°N and 88.02°E coordinates. The bent pipe link is shown in Figure 9. The satellite’s footprint area is 12,896,796 km^2^, whereas the landmass area of study excluding seas is 11,042,891 km^2^. The transmitting and receiving Earth stations are intentionally placed at the boundaries of the satellite footprint to consider the worst-case scenario and the bent pipe communication link is calculated, the 6 GHz uplink C/N = 17.676 dB, the 4 GHz downlink C/N = 13.1123 and composite C/N = 11.8101 dB.

Taking into account the provided satellite footprints, the 5G BS deployment model (macro-BS) was used, based on a large area with urban, suburban, and rural zones and values of (Ds_urb, Ra_urb, Ds_sub, Ra_sub, Rb, Dsru, Rc) = (10, 0.1, 2.4, 0.05, 0.01, 0.003, 0.2). The parameters of each Monte Carlo simulation of interference step randomly generated 5G BS/UE (macro) within the area given above are presented in Table 4 below.

Figure 10, Figure 11 and Figure 12 present the cumulative distribution functions of C/(N+I) for each victim satellite link.

Using values from Table 4 and CDF from Figure 10, Figure 11 and Figure 12, for each of the satellite transponder signal-to-noise degradation was calculated, and the results show that the level degradation is less than 0.1 dB. According to Recommendation ITU-R S.2131 [19], in DVB-S2 link with adaptive modulation and coding C/N reduction by 1 dB equals approx. 10% reduction of spectral efficiency. The obtained results show that in the case of 5G interference, the reduction of the spectral efficiency of satellite links will be very low. The results of signal-to-noise degradation of the considered bent pipe satellite transponders due to the interference from 5G base stations are presented as cumulative distribution functions of C/(N + I) in Figure 13.

When transmitting the signal through satellite communication link errors in received data occur due to the noise. In digital communication, using adaptive modulation with different modulation schemes is instrumental since it is possible to detect and correct errors by adding redundancy which is called forward error correction (FEC). This allows improving spectral efficiency using different code rates by restoring the data without retransmitting it. The more redundant bits are added, the more robust waveform is, but such improvement cost reduction of data rate and latency. The most popular solutions for satellite links are V + RS—convolutional codes with Viterbi decoding and block codes of Reed-Solomon, trellis coded modulation (TCM), TCM + RS—trellis coded modulation with block codes of Reed-Solomon, TPC—turbo product codes, LDPC—block codes with low density of parity checks. In modern satellite models widely used waveforms use phase modulation with different FEC levels. [20,21,22].

Given that the simulation scenarios consider the satellite worst-case link levels where minimum transmit powers are used and the earth stations are located at the boundaries of footprint contours, QPSK and 8PSK modulations would be used in such scenarios. Table 5 contains a theoretical performance of QPSK and 8PSK based on the ETSI EN 302307 DVB-S2 standard.

Knowing *C/N* and link performance of different MODCODs allows us to find an important metric in digital communication—*E_b_/N_o_* which is the normalized signal-to-noise level ratio, the following expression should be used to determine *E_b_/N_o_* [20,21]:(12)$$E_{b}/N_{o}=C/N-10*log\left(\frac{R}{B}\right)$$
where *E_b_/N_o_* the ratio of energy per bit to spectral power density (dB); *N* is noise level in a reference bandwidth (dBW); I is the interference level in a reference bandwidth (dBW); *R* is data rate, (Mbps); *B* is the reference bandwidth (MHz). The values of *E_b_/N_o_* can be used to estimate the bit error rate (BER) of a system. Figure 14 below provides theoretical BER curves of QPSK and 8PSK for AWGN conditions.

For the external interference case when estimating degradation of E_b_/N_o_, the noise level should be presented as a sum of the spectral density of the receiver’s noise and external noise *N_∑_ = N_o_ + I_o_* the levels of *E_b_/(N_o_ + I_o_)* should be checked according to the curves above to calculate BER levels for each MODCOD in Table 4.

## 5. Results

Using the obtained C/N and C/(N + I) value, it is possible to obtain CDF of E_b_/(N_o_ + I_o_) for different MODCODs for Yamal 183E Express 140E and Yamal 90E were derived and presented in Figure 15, Figure 16 and Figure 17.

Comparing the E_b_/N_o_ + I_o_ results with the theoretical QPSK and 8PSK curves in Figure 14, it can be concluded that the BER levels for FSS carriers 8 (183E), 7140E, and 8 (90E) are less than 10^−6^ for QPSK and less than 10^−3^ for 8PSK. The obtained BER levels and their interpretation for each MODCOD are described in detail in further Section 6.

Phased or amplitude modulations are often shown using graphics as a constellation diagram. QPSK and 8PSK modulations have 4 and 8 states respectively and thus are mapped in 4 or 8 positions constellation diagrams. Based on the obtained in our study BER levels constellation diagrams for different modulations can be simulated. Figure 18 shows constellations diagrams of QPSK and 8PSK modulations of satellite demodulators while interfered by 5G base stations. The QPSK constellation diagram depicts BER level 10^−6^, whereas the 8PSK constellation diagram depicts BER level 10^−3^.

## 6. Discussion and Conclusions

The research performed simulations for different orbital position transponders, for each transponder DVB-S2 waveforms with different modulation and coding schemes were taken into account. It was observed that the results depend a lot on the latitudes of 5G base station deployments rather than on the number of base station interferers, the higher latitude led to higher interference due to the lower elevation angles. At the same time, even for low elevation angles, the amount of aggregate interference generated by 5G base stations deployed within victim satellite footprints does not cause harmful interference and the link performance of satellite bent pipe communication transponders does not experience any adverse effect. The analysis was made for two protection criteria. The first one was the signal-to-noise degradation rate which is a general criterion for different satellite systems which shows the reduction of spectral efficiency, according to Recommendation ITU-R S.2131, the tolerable signal-to-noise degradation is 1 dB which corresponds to the 10% of spectral efficiency reduction As may be noted in Figure 13, the signal-to-noise degradation of the satellite with orbital position 90E (presented in Figure 9) is between 0.005 and 0.015 dB, for the satellite with orbital position 140E (presented in Figure 8) it is between 0.04 and 0.055 dB and for the satellite with orbital position 183E (presented in Figure 7) it is between 0.055 and 0.09 dB. Given that the tolerable degradation threshold is 1 dB, it may be seen that the margins are very high and is far less than the tolerable link threshold margin for all three satellite victim receivers ranging between 9.995 dB and 9.91 dB.

The second protection criterion is a more detailed one that takes into account the DVB-S2 standard that satellite systems use within 6/4 GHz transponders and takes into account BER levels for different MODCOD schemes of QPSK and 8PSK that are used to correct errors and recover information. In different communication systems, there are different requirements for BER levels, however overall it may be summarized that a BER level of 10^−9^ or lower is considered as excellent, the BER levels between 10^−9^ and 10^−6^ as very good, the BER levels between 10^−6^ and 10^−3^ as good and BER levels between 10^−3^ and 10^−2^ as acceptable. The BER level of 10^−1^ is considered as bad and higher than 10^0^ as unacceptable. When analyzing link level performance based on the DVB-S2 parameters, it was shown that for the different MODCODs the overall ranges of the BER levels of all three victim receiver satellites for the QPSK modulation did not exceed 10^−6^, and for the 8PSK modulation, did not exceed 10^−3^. For the satellite with orbital position 90E, it can be seen in Figure 17 that for QPSK modulation E_b_/N_o_ + I_o_ levels are between 10.4 dB and 11.4 dB which approximately corresponds to BER level 10^−7^ whereas for 8PSK modulation E_b_/N_o_ + I_o_ levels are between 9.75 dB and 10.42 dB which approximately corresponds to BER level 10^−3^. For the satellite with orbital position 90E, it can be seen in Figure 16 that for QPSK modulation E_b_/N_o_ + I_o_ levels are between 12.2 dB and 13.15 dB which approximately corresponds to BER level 10^−9^ whereas for 8PSK modulation E_b_/N_o_ + I_o_ levels are between 11.5 dB and 12.18 dB which approximately corresponds to BER level 10^−4^. For the satellite with orbital position 90E, it can be seen in Figure 15 that for QPSK modulation E_b_/N_o_ + I_o_ levels are between 11.45 dB and 12.4 dB which approximately corresponds to BER level 10^−8^ whereas for 8PSK modulation E_b_/N_o_ + I_o_ levels are between 10.75 dB and 11.45 dB which approximately corresponds to BER level 10^−3^. These BER levels are acceptable and allow to provide DVB-S2 services. gGiven that the study assumed the worst case when the satellite bent pipe communication link used minimum powers for both uplink and downlink directions, and the earth stations for each transponder were located on the border of the satellite footprints, the obtained results show compatibility since in practice the system won’t operate constantly at minimum powers for such a scenario and BER levels would be even higher.

Additionally, the authors would like to underline that for the 6425–7125 MHz band 5G most likely will have a lower deployment rate compared to the predictions used in the study. These predictions were based on the predictions presented in ITU-R, however previous experience of a similar study in the 26 GHz band showed that in practice deployment rate was much lower compared to the assumptions made in ITU-R. It should be also noted, that the study assumed all countries within the satellite footprint implement 5G in the 6 GHz band, however, while the majority of these countries have plans for 5G in the 6 GHz, in practice not all of them may have plans to utilize this band for 5G, and thus the deployment density may be also lower. Since the study showed compatibility for the highest estimated deployment rate, for the lower deployment rate which will most likely be in practice, the compatibility situation will be even better. Consequently, 5G base stations may be deployed at any region within the services areas of fixed satellite service without any additional restrictions.

## Figures and Tables

**Figure 1 sensors-23-00172-f001:**
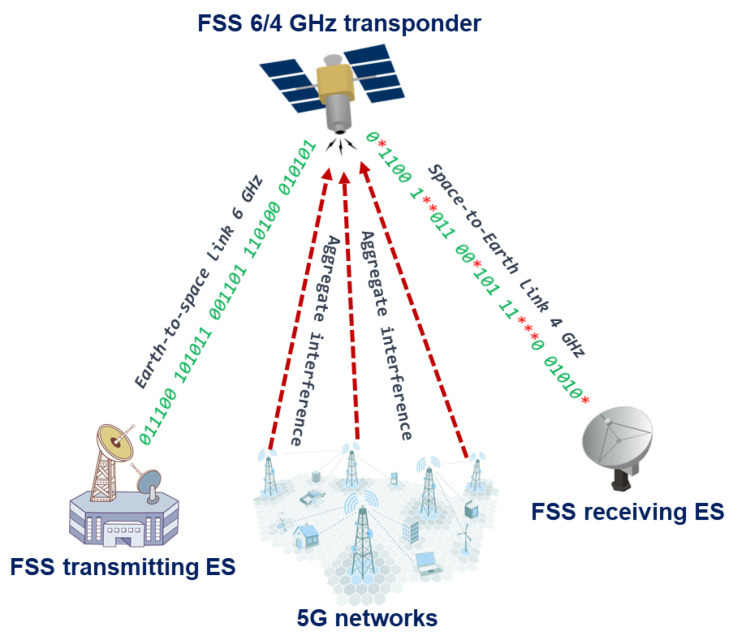
Typical scenario of aggregate interference to the bent pipe satellite link.

**Figure 2 sensors-23-00172-f002:**
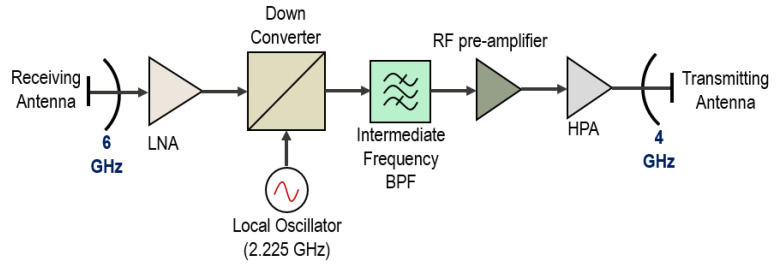
Block diagram of the transponder for 6/4 GHz.

**Figure 3 sensors-23-00172-f003:**
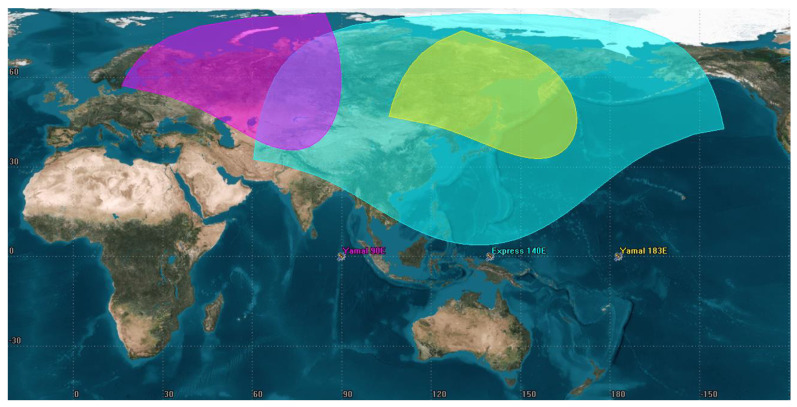
Satellite footprints of the reviewed FSS victim receivers.

**Figure 4 sensors-23-00172-f004:**
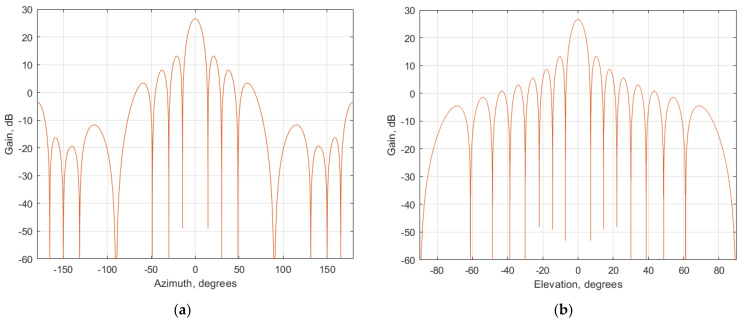
Antenna pattern of urban 5G NR BS (**a**) Horizontal plane; (**b**) Vertical plane.

**Figure 5 sensors-23-00172-f005:**
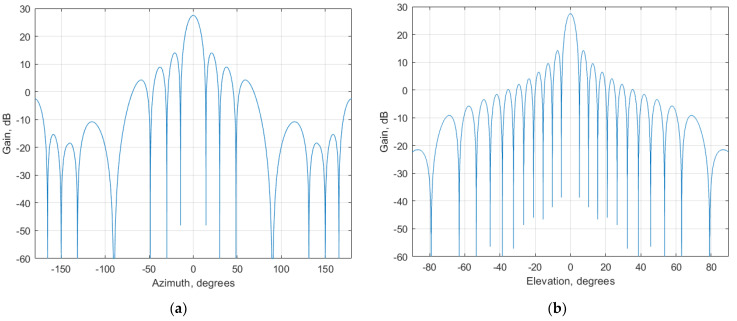
Antenna pattern of suburban 5G NR BS (**a**) Horizontal plane; (**b**) Vertical plane.

**Figure 6 sensors-23-00172-f006:**
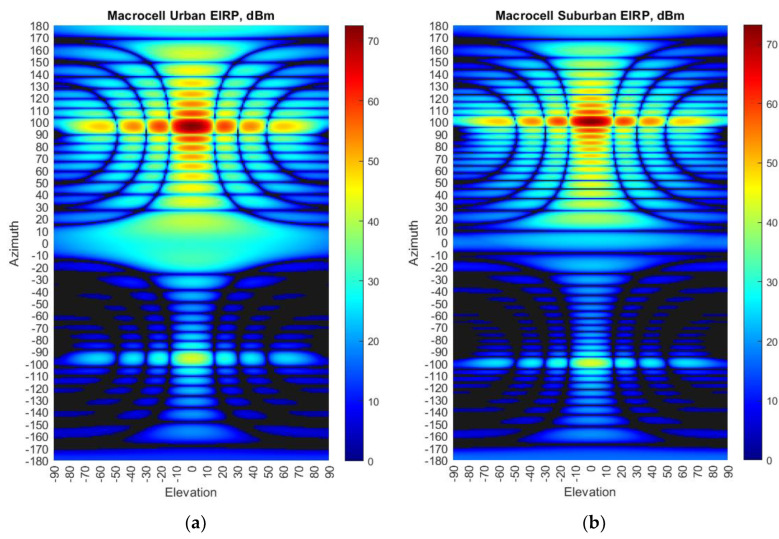
EIRP distribution in azimuth and elevation plane of 5G base stations (**a**) Urban deployment; (**b**) Suburban deployment.

**Figure 7 sensors-23-00172-f007:**
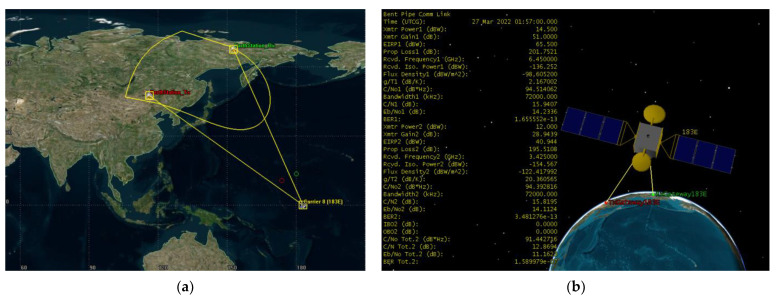
Scenario 1 operation (**a**) 183° E bent pipe transponder operation (**b**) Bent piper of Yamal 183E transponder composite link calculation.

**Figure 8 sensors-23-00172-f008:**
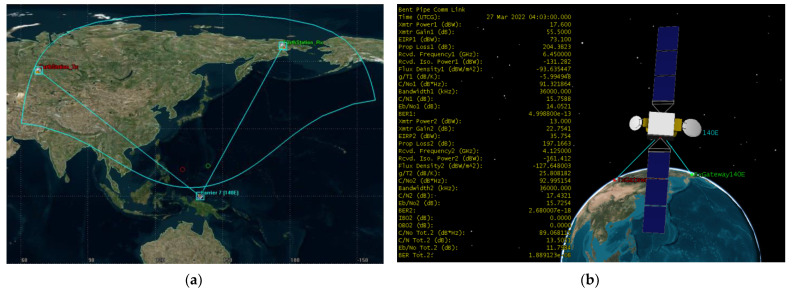
Scenario 1 operation (**a**) 140° E bent pipe transponder operation (**b**) Bent piper of Express 140E transponder composite link calculation.

**Figure 9 sensors-23-00172-f009:**
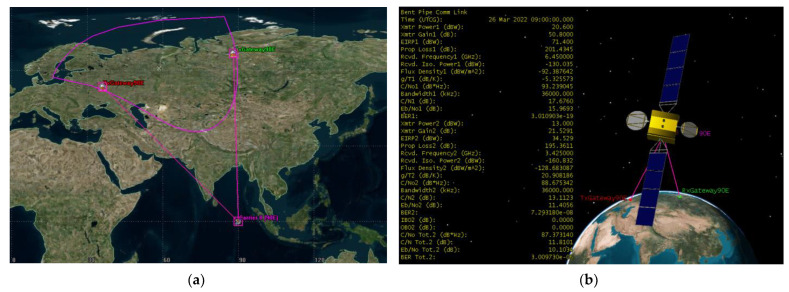
Scenario 1 operation (**a**) 90° E bent pipe transponder operation (**b**) Bent piper of Yamal 90E transponder composite link calculation.

**Figure 10 sensors-23-00172-f010:**
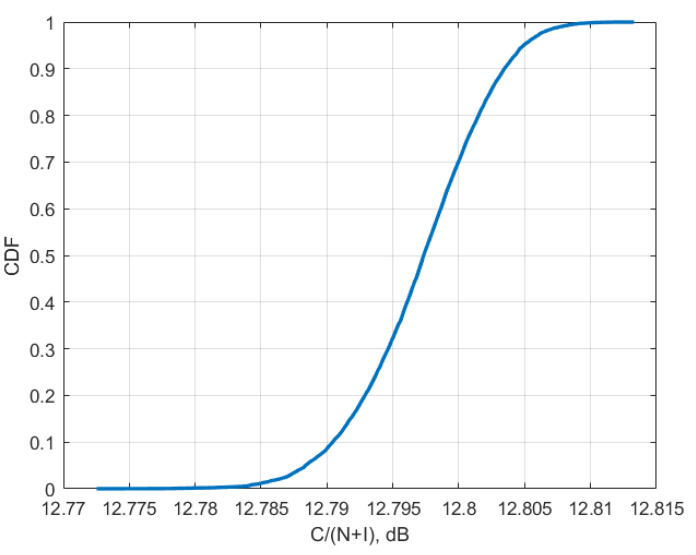
Cumulative distribution function of C/(N + I) for transponder Yama; 183E for 6/4 GHz.

**Figure 11 sensors-23-00172-f011:**
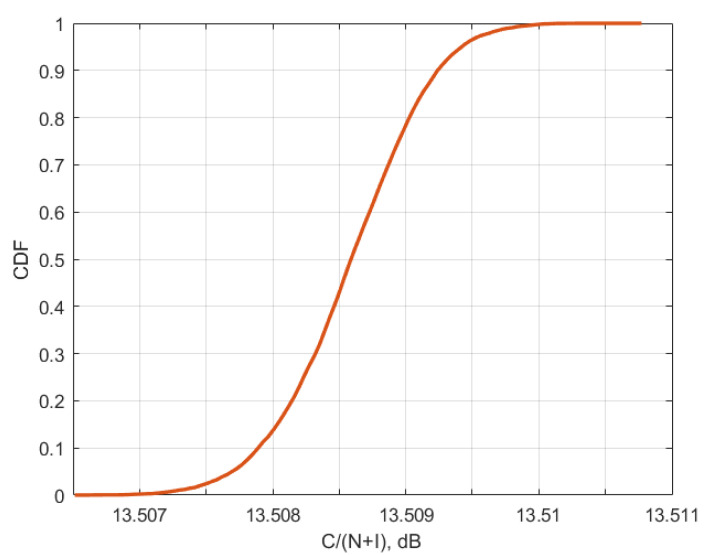
Cumulative distribution function of C/(N + I) for Express 140E for 6/4 GHz.

**Figure 12 sensors-23-00172-f012:**
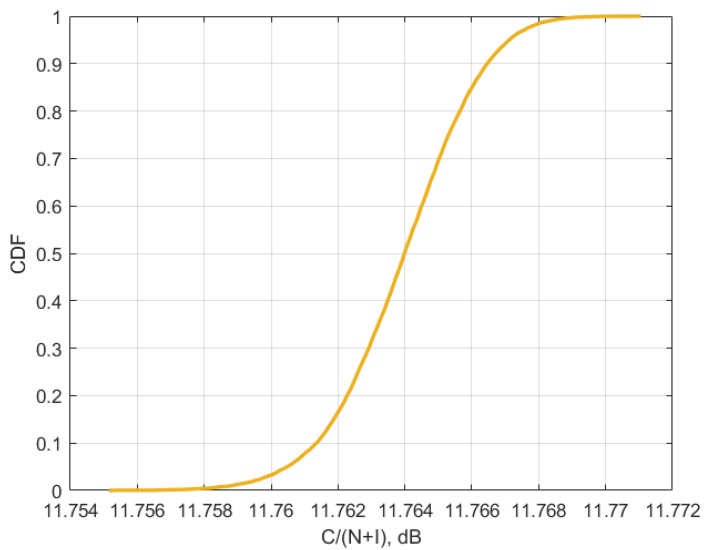
Cumulative distribution function of C/(N + I) for Yamal 90E for 6/4 GHz.

**Figure 13 sensors-23-00172-f013:**
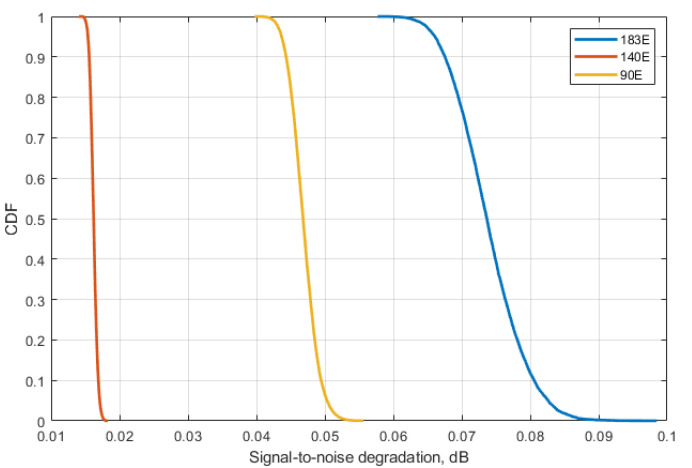
Cumulative distribution functions of C/N reduction for each victim satellite link.

**Figure 14 sensors-23-00172-f014:**
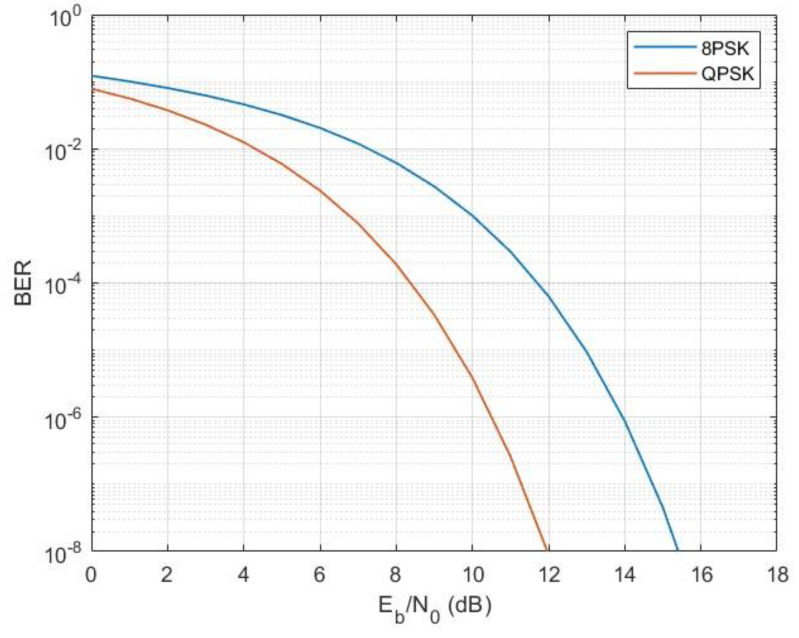
Theoretical curves of BER levels for QPSK and 8PSK.

**Figure 15 sensors-23-00172-f015:**
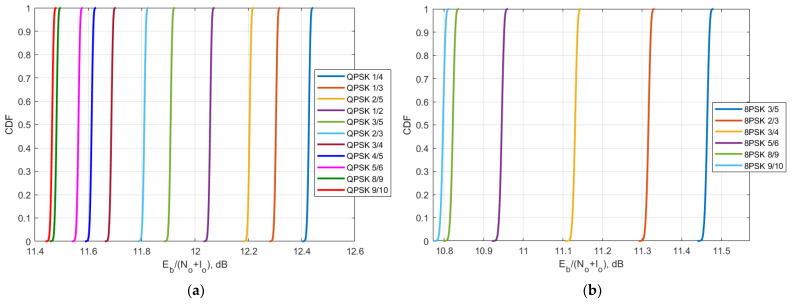
The CDF of E_b_/N_o_ + I_o_ of Yamal 183E: (**a**) for QPSK modulation; (**b**) 8PSK modulation.

**Figure 16 sensors-23-00172-f016:**
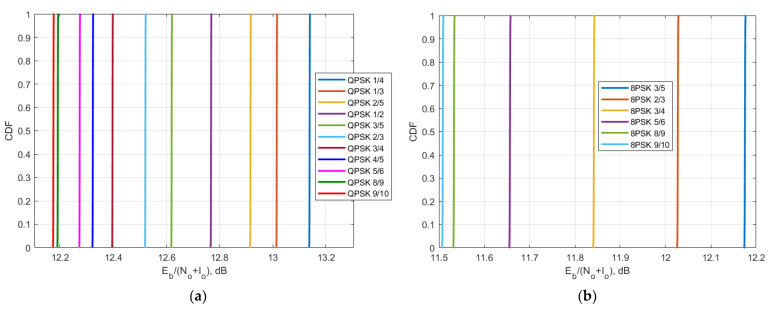
The CDF of E_b_/N_o_ + I_o_ of Express 140E: (**a**) for QPSK modulation; (**b**) 8PSK modulation.

**Figure 17 sensors-23-00172-f017:**
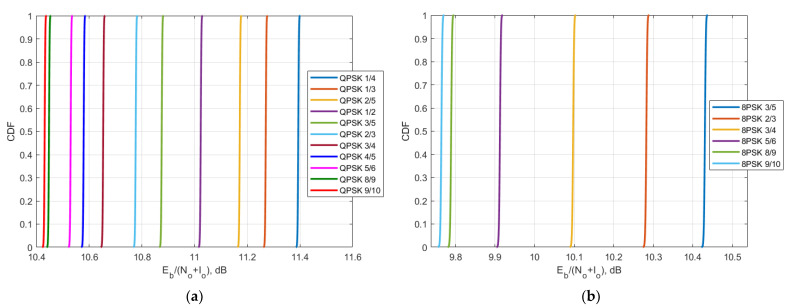
The CDF of E_b_/N_o_ + I_o_ of Yamal 90E: (**a**) for QPSK modulation; (**b**) 8PSK modulation.

**Figure 18 sensors-23-00172-f018:**
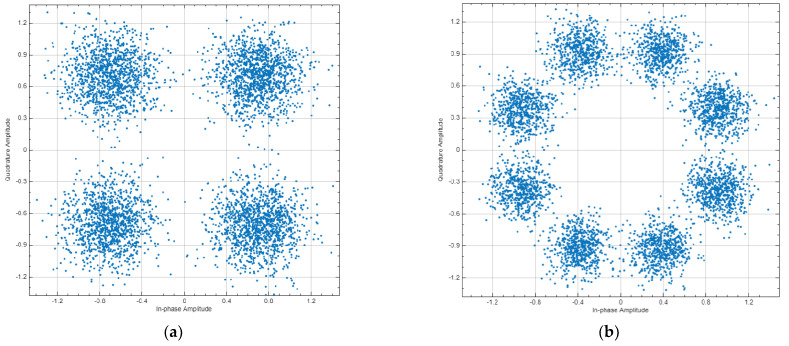
Constellation diagrams of satellite demodulators (**a**) QPSK modulation; (**b**) 8PSK modulation.

**Table 1 sensors-23-00172-t001:** Characteristics of the satellite transponders used in simulations.

Parameter	Yamal 90E	Express 140E	Yamal 183E
Noise bandwidth (MHz)	72	36	72
Peak antenna gain (dBi) (satellite)	32	22	32
Antenna gain pattern beamwidth (satellite)	Rec. ITU-R S.672, LS = −25, Beamwidth: 4.5	Rec. ITU-R S.672, LS = −25, Beamwidth: 14	Rec. ITU-R S.672, LS = −25, Beamwidth: 4.5
Receiver noise temperature (K) (satellite)	500	500	500
Peak transmit antenna gain (dBi) (uplink Earth station)	50.8	55.5	51
Antenna transmit gain pattern (uplink Earth station)	Rec. ITU-R 580-6	Rec. ITU-R 580-6	Rec. ITU-R 580-6
Min power spectral density (dBW/Hz) (uplink Earth station)	−55	−58	−64.1
Min power spectral density (dBW/Hz) (satellite)	−62.6	−62.6	−66.6
Antenna receive gain (dBi) (downlink Earth station)	41.7	46.6	41.5
Antenna receive gain pattern (downlink Earth station)	Rec. ITU-R 580-6	Rec. ITU-R 580-6	Rec. ITU-R 580-6
Receiver noise temperature (K) (downlink Earth station)	120	120	130

**Table 2 sensors-23-00172-t002:** Characteristics of 5G base stations used in simulations.

Parameter	Suburban Macro	Urban Macro
Deployment density (BSs/km^2^)	2.4	10
Antenna height (m)	20	18
Sectorization	3 sectors	3 sectors
Channel bandwidth (MHz)	100	100
Network loading factor (base station load probability)	20%	20%
BS TDD activity factor	75%	75%
Element gain (dBi)	6.4	5.5
Antenna polarization	Linear ± 45°	Linear ± 45°
Antenna array configuration (Column × Row)	16 × 8 elements	16 × 8 elements
Base station horizontal coverage range (degrees)	±60	±60
Base station vertical coverage range (degrees)	90–100	90–100
Mechanical downtilt (degrees)	10	6
Maximum antenna gain (dBi)	27.4	26.5
Conducted power (before Ohmic loss) per antenna element (dBm)	22	22
Conducted power (before Ohmic loss) per antenna per polarization (dBm)	43	43
Conducted power (before Ohmic loss) per antenna (dBm)	46	46
Maximum EIRP density (dBm/100 MHz)	73.4	72.5

**Table 3 sensors-23-00172-t003:** Values for Ra and Rb of 5G for frequency the bands between 6 and 8 GHz.

Parameter	Value
Ra	30% Urban (studied area < 200,000 km^2^)
	10% Urban (studied area > 200,000 km^2^)
	10% Suburban (studied area < 200,000 km^2^)
	5% Suburban (area > 200,000 km^2^)
Rb	2.5% (studied area < 200,000 km^2^)
	2% (200,000–1,000,000 km^2^)
	1% (studied area > 1,000,000 km^2^)

**Table 4 sensors-23-00172-t004:** Parameters of each Monte Carlo simulation step (urban/suburban (rural) macro).

Parameter	Yamal 90E	Express 140E	Yamal 183E
Number of urban 5G BS/Number of active urban 5G BS	68,214/10,232	110,429/16,564	303,050/45,457
Number of suburban 5G BS/Number of active suburban 5G BS	8186/1228	13,252/1988	36,366/5455
Number of rural 5G BS/Number of active rural 5G BS	4254/638	6887/1033	18,901/2835

**Table 5 sensors-23-00172-t005:** Theoretical performance QPSK and 8PSK MODCODs in DVB-S2.

MODCOD	Bandwidth (MHz)	Nyquist Rolloff	Symbol Rate (Msym/s)	Gross Bitrate (Mbps)	Info Bits (Mbps)	Code Bits (Mbps)
QPSK 1/4	36	0.35	26.67	53.33	13.33	40
	72	0.35	53.33	106.67	26.67	80
QPSK 1/3	36	0.35	26.67	53.33	17.78	35.56
	72	0.35	53.33	106.67	35.56	71.11
QPSK 2/5	36	0.35	26.67	53.33	21.33	32
	72	0.35	53.33	106.67	42.67	64
QPSK 1/2	36	0.35	26.67	53.33	26.67	26.67
	72	0.35	53.33	106.67	53.33	53.33
QPSK 3/5	36	0.35	26.67	53.33	32	21.33
	72	0.35	53.33	106.67	64	42.67
QPSK 2/3	36	0.35	26.67	53.33	35.56	17.78
	72	0.35	53.33	106.67	71.11	35.56
QPSK 3/4	36	0.35	26.67	53.33	40	13.33
	72	0.35	53.33	106.67	80	26.67
QPSK 4/5	36	0.35	26.67	53.33	42.67	10.67
	72	0.35	53.33	106.67	85.33	21.33
QPSK 5/6	36	0.35	26.67	53.33	44.44	8.89
	72	0.35	53.33	106.67	88.89	17.78
8PSK 3/5	36	0.35	26.67	80	48	32
	72	0.35	53.33	160	96	64
QPSK 8/9	36	0.35	26.67	53.33	47.41	5.93
	72	0.35	53.33	106.67	94.81	11.85
QPSK 9/10	36	0.35	26.67	53.33	48	5.33
	72	0.35	53.33	106.67	96	10.67
8PSK 2/3	36	0.35	26.67	80	53.33	26.67
	72	0.35	53.33	160	106.67	53.33
8PSK 3/4	36	0.35	26.67	80	60	20
	72	0.35	53.33	160	120	40
8PSK 5/6	36	0.35	26.67	80	66.67	13.33
	72	0.35	53.33	160	133.33	26.67
8PSK 8/9	36	0.35	26.67	80	71.11	8.89
	72	0.35	53.33	160	142.22	17.78
8PSK 9/10	36	0.35	26.67	80	72	8
	72	0.35	53.33	160	144	16

## Data Availability

Not applicable.

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
