# Peer review of "Interference Analysis of 5G NR Base Stations to Fixed Satellite Service Bent-Pipe Transponders in the 6425–7125 MHz Frequency Band"

_sensors, 2022, doi:10.3390/s23010172_

Round 1
Reviewer 1 Report
Q1:In the simulation, only Yamal 90E and 183E orbit satellites and Express 140E orbit satellites were considered as simulation objects of victim receivers. Please explain the reasons for this choice.
Q2:In order for the reader to have a clearer understanding of Figure 6, it is necessary for the author to explain Figure 6 in more detail.
Q3: In the simulation process of Part 4, Scenario 1, Scenario 2, and scenario 3 all have only one set of experimental subjects. In order to make the results more convincing, the authors may consider adding experimental subjects.
Q4: In Part 4, the worst link level of the satellite is considered (starting with line 356). In this scenario, only QPSK and 8PSK modulation methods are selected. It is necessary for the author to explain the reasons in the paper.
Author Response
Q1:In the simulation, only Yamal 90E and 183E orbit satellites and Express 140E orbit satellites were considered as simulation objects of victim receivers. Please explain the reasons for this choice.
Q2:In order for the reader to have a clearer understanding of Figure 6, it is necessary for the author to explain Figure 6 in more detail.
Q3: In the simulation process of Part 4, Scenario 1, Scenario 2, and scenario 3 all have only one set of experimental subjects. In order to make the results more convincing, the authors may consider adding experimental subjects.
Q4: In Part 4, the worst link level of the satellite is considered (starting with line 356). In this scenario, only QPSK and 8PSK modulation methods are selected. It is necessary for the author to explain the reasons in the paper.
Thank you for your questions, the replies are the following:
Q1: The reason for choosing these particular satellites was that they serve higher latitudes, the higher the latitude the lower the elevation angle under which the base stations pointed towards the geostationary arc and thus the interference levels are higher. We wanted to consider the worst-case scenario. So, we’ve chosen the regions with higher latitudes in different parts of the Eurasian continent.
Q2: Thank you for this observation, we have added a more detailed description to the new version of the manuscript.
Q3: This was the typo, C/N levels of the transponders in the text for each scenario are different, we have corrected this in the manuscript. Now they have different sets.
Q4: Thanks for this good question. The reason for choosing these two particular modulation types is that they are the most common for DVB-S2 standard in the 6 GHz band, and the higher order modulations such as 16APSK and 32APSK are more common for Ku bands. Another reason is that the satellites have power regulation, we have considered the situation when the power levels are the minimum possible. For minimum powers, higher modulation orders wouldn’t be possible to be used even without interference, so QPSK and 8PSK would be used for such scenarios. If the satellite would need to use higher order modulations, then the adaptive power control would set different power, but in our view, only minimum power level should be considered to take into account the worst-case. If for the worst case the protection criterion is met, then for the better cases it would be met as well.
Please find the attachment with corrected version, I used reviewer's mode to show the places where changes were made.

Reviewer 2 Report
1. The paper is not well written, there are a lot of typos and grammatical errors.
2. The images are not of good quality, prepare them in 600 dpi
3. The contribution of the paper is not novel and also not depicted clearly.
4. The methodology is not explained properly
5. The results explanation is not properly defined.
6. There are a lot of old references. Follow new papers.
7. The literature survey regarding the work is not done properly.
Author Response
Hello, thank you for your comments. Please see the attachment that contains the replies.

Reviewer 3 Report
1. Authors should provide more detailed information about the simulation environment that they used.
2. Witch simulator did they use to extract Figures 3,7,8,9 ?
3. Authors should explain in more detail the simulation results in paragraph 5 (5. Results). This is important since the conclusions presented in paragraph 5 (line 398, which must corrected to 6) aren’t clearly linked with the presented results in the previews paragraph.
Author Response
Thank you for your questions! Here are the replies below:
- I added that information to the methodology text, MATLAB and STK were used for simulations.
- STK simulator.
- Section 6 was updated with better explanations, please see the attached version with editions.

Round 2
Reviewer 1 Report
I have no further issue for this version.
Reviewer 2 Report
author has incorporated all the changes suggested
Reviewer 3 Report
Authors followed the points that found in the first review.
In my opinion the manuscript can be published in the present form.